# Study on Impact–Echo Response of Concrete Column near the Edge

Yunlin Liu [1], Hongbao Xu [2], Xinxin Ma [3,*], Donghua Wang [2] and Xiao Huang [2]

1 Prefabricated Building Research Institute of Anhui Province, Anhui Jianzhu University, Hefei 230601, China
2 College of Civil Engineering, Anhui Jianzhu University, Hefei 230601, China
3 Graduate School, Anhui Jianzhu University, Hefei 230601, China
* Correspondence: maxinxin@ahjzu.edu.cn

**Abstract:** The impact–echo method is a superior method for detecting the health of concrete structures, but it has the disadvantage of significant errors when identifying structural boundaries. In order to overcome this limitation, this paper proposes a calculation method using a cross-sectional vibration mode in combination with the impact–echo method to detect concrete columns. The variation of the predominant frequency in the mid-column region and the critical boundary is studied. The influence of the edge on the detection results is analyzed. The differences in eigenvalues under different cross-section vibration modes are revealed. A quantitative method for evaluating concrete health using the impact–echo method is further established. Through field tests and finite element simulation calculations, it was verified that the eigenvalues in the fixed mode are very consistent with the predominant frequency measured near the edge region. This makes up for the defect of inaccurate measurements when the impact–echo method is used to detect the edge area of concrete members. The impact–echo method can be better used for the non-destructive testing of concrete members.

**Keywords:** impact–echo; finite element simulation; dominant frequency; column edge

## 1. Introduction

When installing a frame column at a construction site, the reserved steel bar between the frame column and the floor may be uncompacted during the connection, resulting in delamination defects, which in turn affect the overall safety performance of the building [1,2]. Among the non-destructive testing techniques commonly used to evaluate the structural conditions of concrete, the impact–echo (IE) method is an effective method for detecting the internal working conditions of the concrete, the thickness of the steel protective layer, the thickness of the slab, and the bonding conditions of the lower interface. When the internal acoustic impedance of the concrete component is uniform, the component can be regarded as a linear elastic, isotropic medium that will produce tensile and compressive deformation. The IE method can detect concrete elements according to the change of frequency, and the fundamental frequency obtained by vibration is converted into the thickness of the member by the equation [3–8]. However, in the actual application process, the detection results of the impact–echo method are greatly affected by the boundary conditions. Therefore, this paper mainly studies the influence of the edge of the measured object on the impact–echo without defects and material degradation.

Pospisil, K., has proven that the impact–echo method is more sensitive to reinforcement conditions. The dominant frequency displacement obtained through fast Fourier transform corresponds to the change of steel bars in concrete beams. Studies have shown that steel damage or fracture affects the predominant frequency. Therefore, in the finite element simulation, the influence of the steel bar on the model needs to be considered [9]. Kee, S.H., et al. proposed in their study that the water or harmful materials in concrete

causes various durability problems by reducing the stiffness of concrete. In addition, they can cause certain defects or delaminations in concrete members. These factors affect the accuracy of the impact–echo method. Therefore, when designing the test, it is necessary to supervise the site to ensure that the test components meet the required standards during the pouring process and eliminate the possibility of internal defects [10]. Xiangyang Xu analyzed a series of high-order vibration modes of thin-walled beams. Analyzing high-order vibration modes in these structures requires more accurate and precise methods. The overall vibration modes related to the deformation of rigid sections, such as bending and torsion, can be detected through classical and shear refinement theories [11,12]. Lee, C., proposed that in practice, vibration sensors (such as displacement sensors, accelerometers, and microphones) are located near the tapping point to measure the dynamic response of the object. The measured transient time signal reveals the maximum (peaks) of certain frequencies dominated by non-propagating waves (or resonance modes) via Fourier transform (amplitude spectrum). The interpretation of non-propagating waves is an important process in IE data analysis. The fundamental thickness stretch mode normally dominates the spectral response of a plate if there are no near-surface defects, such as cracking, debonding, and honeycombs, in the concrete [13]. In the process of test and simulation detection, the predominant frequency detected in the middle of the column is the eigenvalue of the fundamental thickness stretch mode of the column.

When the impact–echo method is used to detect the concrete columns, the predominant frequency within a specific range from the edge has a low-frequency offset. In this paper, the two-dimensional cross-section of the concrete column is simulated using the finite element method to determine whether the predominant frequency under the influence of boundary conditions was related to the eigenvalue change of the cross-section vibration mode [14–19]. By analyzing the signals collected at the boundary of the concrete column, it was found that the decrease in the dominant frequency was related to the torsional deformation of the column. Currently, the sensor does not identify the thickness frequency of the concrete column but the eigenvalue under the fixed mode.

## 2. Background

### 2.1. Principle of Impact–Echo Method

The impact-echo method is a non-destructive testing method that uses transient impact to excite low-frequency stress waves to propagate the inside of the object to be tested. The generated P and S waves propagate along the hemisphere wavefront inside the structure, and the R wave propagates on the surface of the structure to be measured. During the propagation of stress waves, reflections occur when significant defects are encountered in the medium [20]. The transient resonance caused by reflection changes the displacement of the surface of the structure to be measured, and then the signal generated by the resonance is received by the sensor near the impact point. Usually, the signal that the sensor is able to receive is generated by P waves. The displacement waveform signal with periodic property is obtained by signal amplification, and then the time-displacement signal can be converted into amplitude spectra via fast Fourier transform (FFT) [21]. The dominant frequency in the amplitude spectra reflects the thickness or internal defect of the structure. The specific location of the thickness or defect of the structure can be calculated using the following formula [22,23].

$$f = \frac{\beta k}{n} \frac{C_P}{T} \tag{1}$$

where

$$C_P = \sqrt{\frac{E(1-\nu)}{\rho(1+\nu)(1-2\nu)}} \tag{2}$$

$k$ is the geometric correction coefficient; $T$ is the cross-sectional thickness of the structure; $n$ is a constant that depends on the acoustic impedance and generally takes 2 or 4 [24];

$\beta$ is a correction coefficient with a value range of 0.75~0.96; $C_P$ is the longitudinal wave velocity; and $E$, $\rho$, and $\nu$ are the elastic modulus, density, and Poisson's ratio of the test object, respectively [25,26].

### 2.2. Experimental Verification

This section examines the impact–echo test of the concrete column carried out via a field test to verify the phenomenon of the deviation of the predominant frequency and determines the influence range of the boundary to provide experimental support for the subsequent numerical simulation.

The boundary region was detected using SPC-MATS and impact–echo instruments, as shown in Figure 1, respectively. The test was based on the theory of stress wave propagation. During the test, the excitation signal was generated at the excitation point of the steel ball with a diameter of 17 mm. The sensor received the signal of the surface particle vibration, amplified it in the signal amplifier, and transmitted it to the instrument host. The analog signal was converted into a digital signal. The received signal was processed by analytical software to obtain the predominant frequency of the impact signal at different distances from the boundary. During the test, the acquisition time of the instrument was 8.192 ms, the sampling interval was 2 μs, the number of acquisitions was 4096, and the acquisition frequency was set to 500 kHz. The test instruments used the data acquisition system to receive the voltage signal generated by the vibration of the small steel ball on the surface of the component to be tested by the sensor. Therefore, the sensor should be in complete and stable contact with the surface of the component to be tested.

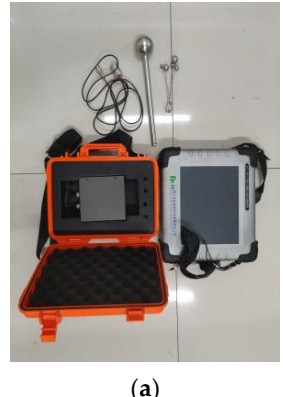
(**a**)

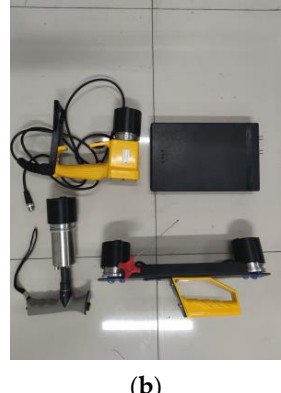
(**b**)

**Figure 1.** (**a**) SPC-MATS instrument; (**b**) impact–echo instrument.

In this paper, in order to facilitate the study of the frequency offset phenomenon of concrete columns under boundary conditions, the band tape was used to detect the line at the edge. A survey line was arranged at the edge every 1 cm from the middle of the column to the edge. The direction of the survey line was perpendicular to the ground, and each survey line was numbered in turn using numbers 1, 2, and so on, as shown in Figure 2. The concrete density was 2400 kg/m$^3$, the elastic modulus was $3.25 \times 10^{10}$ Pa, and the Poisson's ratio was 0.2 (The concrete strength grade is C40). Since the test site was at a construction site, both ends of the column were fixed between the floor and the foundation.

### 2.3. Numerical Models

Because in the actual detection process, the knock position, the sensor placement position, and the intensity of each knock are slightly different, it affects the received signal. Therefore, it is necessary to carry out numerical analysis to understand the transient impact response of the concrete column. The strain generated by the shock wave on the concrete surface is deficient. Therefore, the linear elastic material model suits concrete with low strain levels. In the impact–echo test, the excitation of the stress wave is generated by a small steel ball (17 mm in diameter) falling onto the surface of the concrete. In this paper,

to accurately simulate the actual impact load, ABAQUS/Explicit was used to simulate and analyze the transient impact process of a steel ball falling from a certain height on the surface of a concrete column (in the simulation, the falling process of a steel ball was simplified as a semi-periodic harmonic force [27]). The displacement of particles on the surface of concrete members was collected near the impact point so as to obtain the propagation information of stress waves in concrete members. On this basis, the time domain diagram of the harmonic motion was transformed into a frequency spectrum diagram via fast Fourier transform (FFT) for specific analysis. In this paper, the response of stress waves under the boundary condition of the concrete column is studied and the influence of boundary conditions on the signal is discussed. The ABAQUS finite element simulation was used to simulate the signal received by the signal sensor when the steel ball was knocked near the boundary of the concrete column, then the signal was extracted by Origin software to perform Fourier transform in order to obtain the spectrum diagram, and the change in its predominant frequency was analyzed. The impact form is shown in Formula (3).

$$F(t) = \begin{cases} F_{max}\sin\left(\frac{\pi t}{t_c}\right) & 0 \le t \le t_c \\ 0 & t > t_c \end{cases} \tag{3}$$

The maximum simulated concentrated force was 30 N. The duration of shock action was calculated by the formula (12–74 μs) [28,29]. The signal was collected every $2 \times 10^{-6}$ s, and the corresponding sampling frequency was 500 kHz. The maximum effective frequency generated by the small steel ball was about 17.1 kHz because the density of concrete was 2400 kg/m³, the elastic modulus was $3.25 \times 10^{10}$ Pa, Poisson's ratio was 0.2, the wave velocity calculated by Formula (2) was 4000 m/s, and the wavelength was approximately 23.39 cm when calculated. Then, according to Formula (4), the maximum mesh size was estimated to be 10 mm. The unit type was C3D8R, where C is the entity, 3D represents three dimensions, 8 is the number of nodes of the unit, R is the reduced integral unit, and the grid is divided by sweeping.

$$L_{min} \le \frac{\lambda_{min}}{20} \tag{4}$$

The end surfaces of concrete columns often cause many frequency peaks to appear in the spectra of the simulation results, and this is because a reflection phenomenon occurs when the stress wave reaches the end surface of the concrete column during the propagation process. This paper creates an infinite element on the end surface of the concrete column, extending the column's end surface infinitely so that the waves inside the member are not affected by the influence of the end surface. At the edge, 5 cm is used as the node for impact, from the middle of the column to the boundary, and 5 cm above the impact point is used as the signal receiving point. When entering the mutation range, the node accurately arranges an impact point for each 1 cm. The model is shown in Figure 3.

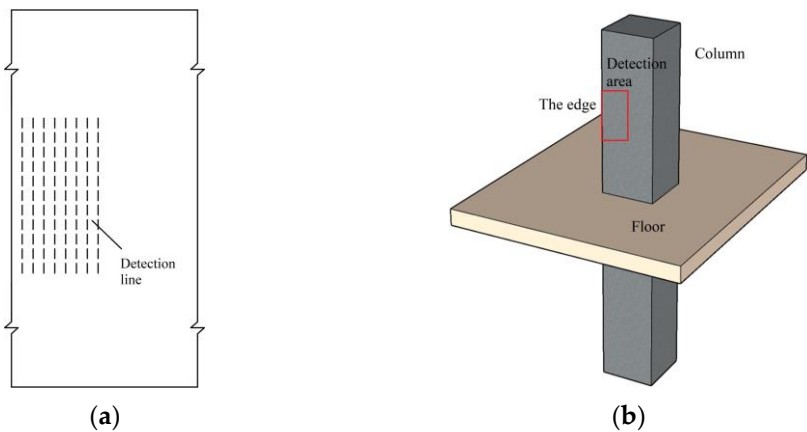

**Figure 2.** (**a**) details of the detection area; (**b**) field testing at the boundary.

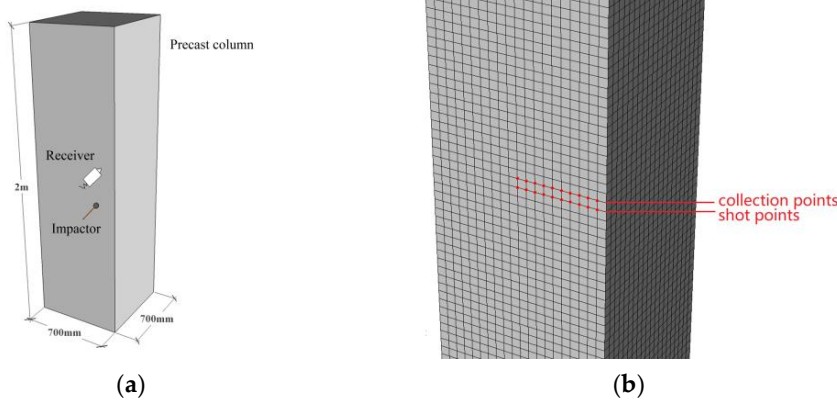

(a)  (b)

**Figure 3.** (**a**) Numerical simulation diagram; (**b**) line layout diagram.

## 3. Impact Response of Concrete Column

### 3.1. Experimental Result

In the experiment, the time history curve data obtained in the middle of the column and the data obtained at the boundary were Fourier transformed by the computer to obtain the dominant frequency corresponding to the column and the edge. The specific results are shown in Figure 4:

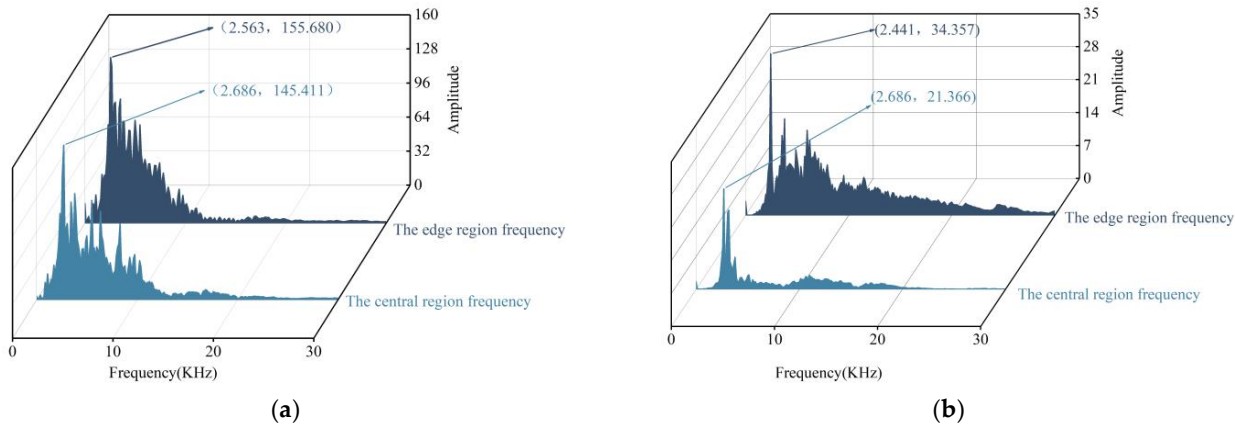

(a)  (b)

**Figure 4.** (**a**) Impact–echo spectrogram; (**b**) SPC-MATS spectrogram.

Table 1 shows the test results of concrete columns via the SPC-MATS and impact–echo methods, respectively. After analyzing the collected signal, the typical dominant frequencies in the influence range of the column and the boundary were compared. From the analysis of the test results, it can be seen that the predominant frequency in the middle of the concrete column is 2.686 kHz, and the dominant frequencies at the edge are 2.563 kHz (impact–echo) and 2.441 kHz (SPC-MATS). When the two instruments detect the concrete column's edge area, the predominant frequency's value decreases. As the acquisition point is closer to the central area of the column when the acquisition point crosses a specific node, the predominant frequency mutates. This is because when the collection point is located in the edge area of the column, the tapping causes the column to undergo torsional deformation. When the collection point is closer to the middle area of the column, it is converted to the bending deformation of the column. The sensor records different modal frequencies.

Through the above test results, it can be seen that the edge causes inevitable interference in impact–echo detection. When the acquisition point is located in the edge area, the predominant frequency corresponds to the dominant frequency generated by torsional deformation. When the acquisition point is located in the middle of the column, the predominant frequency corresponds to the dominant frequency generated by

bending deformation. In order to determine the influence range of the edge on the predominant frequency, it is necessary to further use the finite element software to simulate each working condition.

**Table 1.** Comparison of typical dominant frequencies of the edge test results.

| Edge Distance (cm) | 1 | 10 | 19 | 20 | The Middle of the Column |
|---|---|---|---|---|---|
| Impact–echo measured value (kHz) | - | 2.563 | - | 2.686 | 2.686 |
| SPC-MATS measured value (kHz) | 2.441 | 2.441 | 2.686 | 2.686 | 2.686 |

### 3.2. Edge Simulation Results

In order to study the transient response of concrete frame columns under point impact, a three-dimensional finite element simulation was carried out. During the analysis, a dynamic load was applied at the edge of the column and the predominant frequency obtained by the analysis at the receiving point was compared with the predominant frequency in the central region of the column.

The time history curve and predominant frequency of the central region of the column obtained were via simulation. The results are shown in Figure 5:

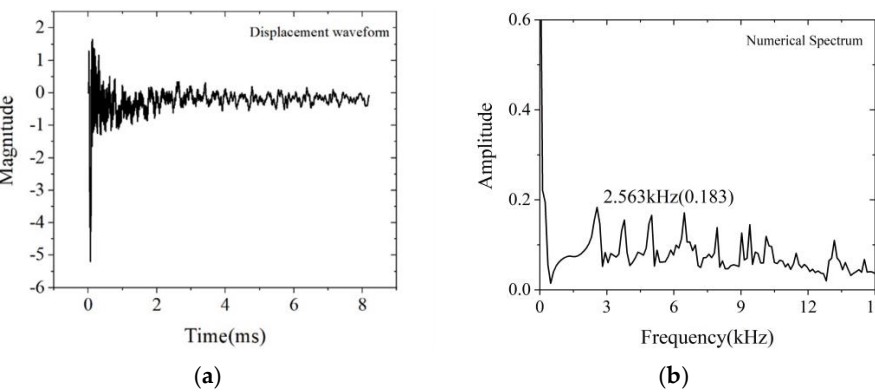

(a)          (b)

**Figure 5.** Test in the middle of the column under the edge condition: (**a**) time−displacement curve; (**b**) amplitude spectra.

As can be seen from Figure 4, the simulated predominant frequency of the middle part of the concrete is 2.563 kHz. To further study the influence of the edge on the dominant frequency, the acquisition points were set up every 5 cm from the column to the edge for the application of the load and the acquisition of the signal, and the results are shown in Figure 6:

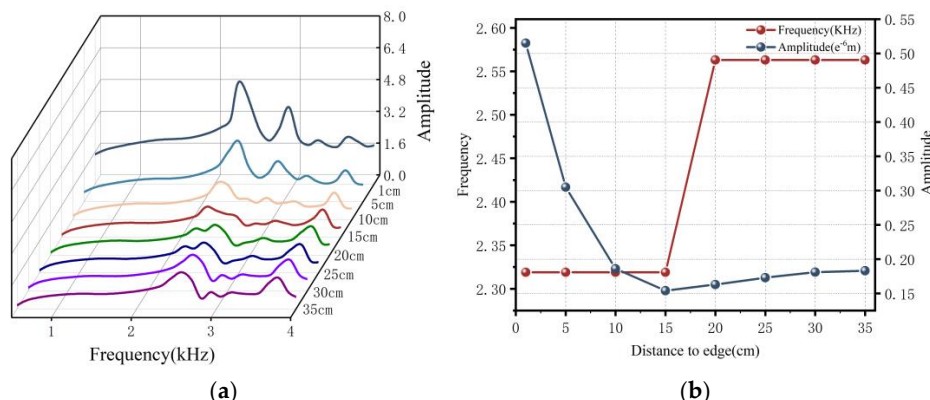

(a)          (b)

**Figure 6.** The edge acquires a spectrum every 5 cm: (**a**) spectrogram; (**b**) frequency change diagram for each point.

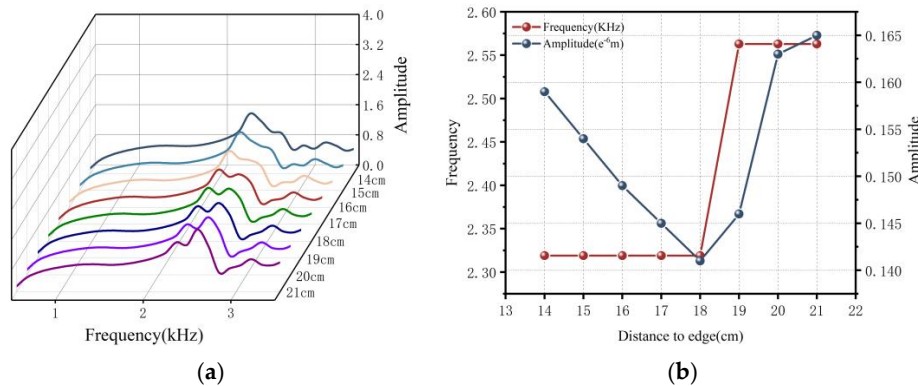

**Figure 7.** The edge acquires a spectrum every 1 cm: (**a**) spectrogram; (**b**) frequency change diagram of each point.

The simulation position is impacted at different positions from the edge, and the spectrum analysis results are as follows:

Table 2 shows that the simulation results are divided into two parts after analysis. In the range of 1–15 cm from the edge, the predominant frequency of the signal acquisition point was 2.319 kHz, which was lower than the predominant frequency of the central region of the column. When the distance from the edge is 20–35 cm, the predominant frequency of the signal acquisition point changed from 2.319 kHz to 2.563 kHz and then remained unchanged. Interestingly, the amplitude corresponding to the dominant frequency also changed. This was because the column was subjected to torsional deformation at the edge. At this point, the impact energy was only absorbed by the column angle, the bending deformation was large, and the amplitude was at its highest level. If the tapping point is located in the middle of the column, the mass of the column absorbs the impact energy and the bending deformation remains small, so the amplitude is low. Therefore, when the predominant frequency is at a low frequency, the amplitude decreases with the increase in distance. When the striking point is located in the middle of the column, the amplitude increases with the distance. Preliminary simulation showed that the frequency mutation was located 15–20 cm from the edge. Combined with Table 3, the variation of predominant frequency and amplitude in this range was analyzed. When the distance from the edge was 14~18 cm, the predominant frequency of the signal acquisition point was still low. When the distance from the edge was 19~21 cm, the frequency of the signal acquisition point became the dominant frequency in the middle of the column. At the same time, the amplitude corresponding to the main frequency also changed. Based on the above two tables, it can be seen that for 700 × 700 mm concrete columns, when the impact–echo method is used to detect at 19 cm from the edge, the predominant frequency changes, and the amplitude carried by the low frequency gradually decreases with the distance from the edge. The amplitude accepted by the high frequency increases with the distance from the edge.

**Table 2.** The edge collects signal analysis results from every 5 cm.

| Distance (cm) | 1 | 5 | 10 | 15 | 20 | 25 | 30 | 35 |
| --- | --- | --- | --- | --- | --- | --- | --- | --- |
| Frequency (kHz) | 2.319 | 2.319 | 2.319 | 2.319 | 2.563 | 2.563 | 2.563 | 2.563 |
| Amplitude ($10^{-6}$ m) | 0.515 | 0.305 | 0.186 | 0.154 | 0.163 | 0.173 | 0.181 | 0.183 |

**Table 3.** The edge collects signal analysis results from every 1 cm.

| Distance (cm) | 14 | 15 | 16 | 17 | 18 | 19 | 20 | 21 |
| --- | --- | --- | --- | --- | --- | --- | --- | --- |
| Frequency (kHz) | 2.319 | 2.319 | 2.319 | 2.319 | 2.319 | 2.563 | 2.563 | 2.563 |
| Amplitude ($10^{-6}$ m) | 0.159 | 0.154 | 0.149 | 0.145 | 0.141 | 0.146 | 0.163 | 0.165 |

### 3.3. Cross-Sectional Mode

The above experimental results show that the measurement accuracy is biased when the impact–echo is within the influence range of boundary conditions. This is because the transient response excited by the impact–echo is composed of multiple resonance frequencies caused by the cross-sectional vibration mode. The echo response from the concrete column is mainly controlled by the wave propagating in its cross-section. Various reflections and interferences produce cross-sectional vibration modes with different frequencies. The sensor accepts the natural frequency excited by the ball in different modes. During the test, the experimenter has different frequencies received by the sensor because of the sensor's different placement and the impact force of the hammer on the measured object. At this point, the excited wave is also affected by the boundary, resulting in a deviation between the measurement accuracy and the amplitude. The corresponding vibration mode is excited when the specified frequency is one of the structure's natural frequencies.

It has been shown that the response of a rod subjected to a short-term transverse impact consists of frequencies corresponding to the cross-sectional vibration pattern of the rod [30,31]. In the case of concrete columns, it is shown that the boundary response also consists of multiple natural frequencies associated with the vibration pattern. The eigenvalue analysis of the two-dimensional finite element model of the concrete column were used to obtain the vibration mode and the corresponding natural frequency. The plane strain finite element model can be used to solve the cross-section mode shape and the corresponding eigenvalues. In the case of two-dimensional cross-section concrete columns, the signal response is also composed of multiple natural frequencies related to the vibration mode. The numerical model uses linear elastic and isotropic material properties to simulate the working conditions of the cross-section of the concrete column. When there are steel bars in the model, the measured dominant frequency is the same as that of the dense concrete with the same thickness. The reflection frequency generated by the interface between concrete and steel bars cannot be identified. This phenomenon is mainly due to the limited interface of the steel bars and the curved shape, resulting in weak echo reflection intensity. Therefore, in the finite element simulation, the influence of steel bars is small and can be ignored, except for large-diameter steel bars and exposed bars. The density, elastic modulus, and Poisson's ratio were 2360 kg/m$^3$, 30,000 MPa, and 0.2, respectively. The longitudinal wave velocity was 4000 m/s. The following is the analysis results of the cross-sectional eigenvalues of concrete columns.

Figure 8 shows the shapes and eigenvalues of the two main cross-sectional modes of concrete columns. The shape coefficient for a square cross-section was usually 0.87, and n was 2. The predominant frequency calculated by the Formula (1) was 2.5 kHz. Consistent with the simulation results and compatible with the modal two eigenvalues. It can be seen that when the acquisition area is located at the center of the column, the predominant frequency obtained by the analysis is the eigenvalue corresponding to Mode 2 in the cross-section vibration mode. At this point, wave propagation is not affected by the boundary.

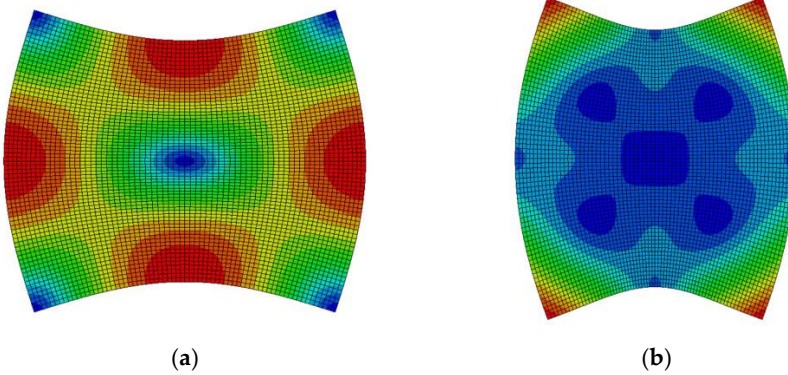

(**a**)          (**b**)

**Figure 8.** Sectional vibration mode and eigenvalue. (**a**) Mode 1 (*f* = 2324 kHz); (**b**) Mode 2 (*f* = 2540.2 kHz).

When the acquisition area is located in the influence range of the edge, the finite element simulation result is 2.391 kHz, which conforms to the eigenvalue corresponding to mode one. According to the above research, the predominant frequency in the middle of the column, the second eigenvalue of the mode, is taken as the primary frequency $f_0$. The eigenvalue of the first mode is $0.9 f_0$.

## 4. Result Validation

The column size in the above test and simulation was $700 \times 700$ mm. In order to verify that the reduction of predominant frequency is a common phenomenon in the edge region of concrete columns and to determine the relationship between the region's influence range and the column's size, concrete columns with cross-section sizes of $400 \times 400$ mm and $1000 \times 1000$ mm were simulated and calculated. Table 4 is the detection data of the $400 \times 400$ mm column. The frequency change trend is shown in Figure 9.

**Table 4.** The edge collects signal analysis results from every 1 cm ($400 \times 400$ mm).

| Distance (cm) | 4 | 5 | 6 | 7 | 8 | 9 | 10 | 11 |
|---|---|---|---|---|---|---|---|---|
| Frequency (kHz) | 4.027 | 4.027 | 4.027 | 4.027 | 4.027 | 4.515 | 4.515 | 4.515 |
| Amplitude ($10^{-6}$ m) | 0.420 | 0.397 | 0.374 | 0.352 | 0.328 | 0.311 | 0.318 | 0.326 |

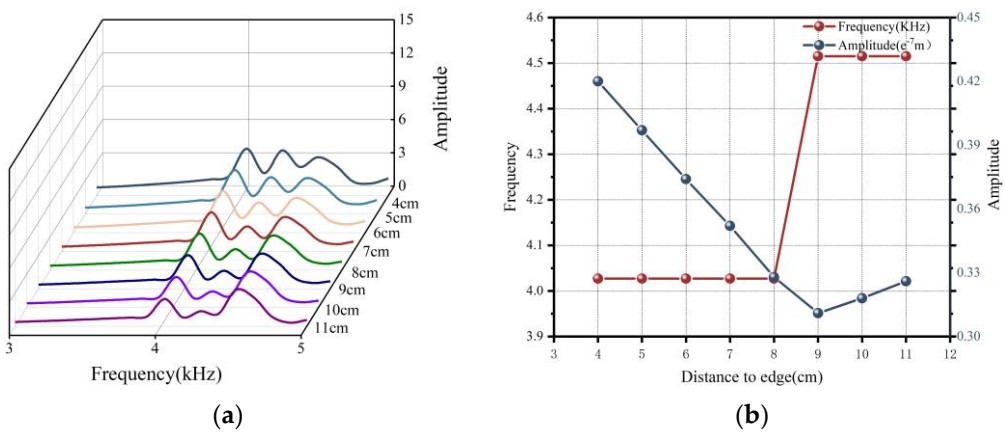

(**a**)        (**b**)

**Figure 9.** The edge acquires a spectrum every 1 cm: (**a**) spectrogram; (**b**) frequency change diagram of each point.

According to the analysis results of the receiving point signal, when the impact is loaded at each interval of 1 cm in the range of the $400 \times 400$ mm column distance edge mutation, the predominant analytical frequency of the signal at 4–8 cm from the edge is 4.027 kHz and the predominant analytical frequency of the signal at 9–11 cm from the edge is 4.515 kHz. The mutation point is 9 cm from the edge. Within this range, the amplitude of frequency 4.027 kHz gradually decays from $0.420 \times 10^{-6}$ mm to $0.328 \times 10^{-6}$ m, and the amplitude of frequency 4.515 kHz gradually increases from $0.311 \times 10^{-6}$ m to $0.326 \times 10^{-6}$ m. It can be seen that the frequency 4.027 kHz offset range of the $400 \times 400$ mm column is 1–8 cm from the edge, and the amplitude carried by the frequency gradually decreases with the increase in the distance from the edge. The frequency amplitude of 4.515 kHz gradually increases with the distance from the edge, and finally, the predominant frequency tends to be stable.

Table 5 is the detection data of the $1000 \times 1000$ mm column. The frequency change trend is shown in Figure 10.

When the $1000 \times 1000$ mm column is impacted at each interval of 1 cm from the edge mutation range, the predominant analytical frequency of the signal at the distance of 24–27 cm from the edge is 1.709 kHz, the predominant analytical frequency of the signal at the distance of 28–31 cm from the edge is 1.831 kHz. The mutation point is 28 cm from the edge. In this range, the amplitude carried by the low-frequency 1.709 kHz gradually

decreases from $0.145 \times 10^{-6}$ m to $0.135 \times 10^{-6}$ m, and the amplitude carried by frequency 1.831 kHz gradually increases from $0.132 \times 10^{-6}$ m to $0.136 \times 10^{-6}$ m. It can be seen that frequency 1.709 kHz offset range of the $1000 \times 1000$ mm column is 1–27 cm from the edge, and the amplitude carried by frequency gradually decreases with the increase in the distance from the edge. The amplitude carried by frequency 1.831 kHz gradually increases with the distance from the edge, and finally, the predominant frequency tends to be stable. Figure 11 shows the edge influence range of two different sizes of concrete columns.

**Table 5.** The edge collects signal analysis results from every 1 cm ($1000 \times 1000$ mm).

| Distance (cm) | 24 | 25 | 26 | 27 | 28 | 29 | 30 | 31 |
|---|---|---|---|---|---|---|---|---|
| Frequency (kHz) | 1.709 | 1.709 | 1.709 | 1.709 | 1.831 | 1.831 | 1.831 | 1.831 |
| Amplitude ($10^{-6}$ m) | 0.145 | 0.143 | 0.139 | 0.135 | 0.132 | 0.133 | 0.135 | 0.136 |

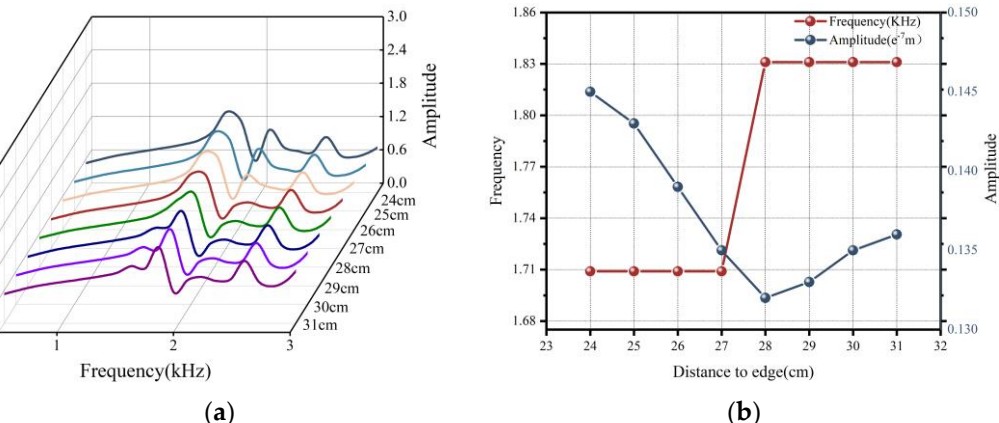

(**a**)    (**b**)

**Figure 10.** The edge acquires a spectrum every 1 cm: (**a**) spectrogram; (**b**) frequency change diagram of each point.

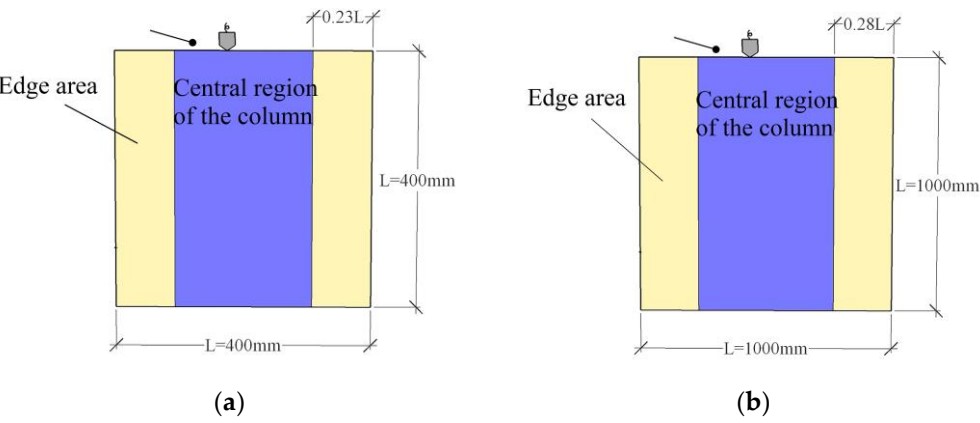

(**a**)    (**b**)

**Figure 11.** The influence range of edge area: (**a**) $400 \times 400$ mm; (**b**) $1000 \times 1000$ mm.

According to the simulation of the above three different section sizes of concrete columns, the influence range of the edge region can be judged. When the striking point is located within the scope of the edge influence distance, the predominant frequency of the receiving point signal produces a low-frequency offset phenomenon within this distance, which will cause errors in the actual test. The frequency mutation occurs at a particular position as the distance between the striking point and the edge moves further away. Then, the predominant frequency tends to be stable. For concrete columns with different widths, when the tester uses the impact–echo method to detect the width of the concrete column in the range of 0.23 to 0.28 times, the phenomenon of predominant frequency reduction

occurs. An actual field test should avoid a range detection of 0.3 times thickness, as at this point, it no longer causes the phenomenon of predominant frequency reduction.

## 5. Conclusions

According to the research, when the impact–echo method is used to detect concrete columns, the analytically obtained predominant frequency is affected by the edge and the frequency value is reduced. In this paper, to determine the influence range of the edge, the method of field test combined with numerical simulation was used to analyze the frequency under relevant working conditions. The vibration mode of the square cross-section was simulated using the plane strain finite element model, and the natural frequency of each mode was calculated to further explain the influence of the boundary on the predominant frequency. Through the above research, the main conclusions are as follows:

(1) When the impact point is located in the edge region's influence range, the predominant frequency's value decreases. When the impact point is farther away from the edge, the predominant frequency returns to the normal level and the predominant frequency is related to the width of the concrete column.

(2) When the impact–echo method is used in the actual detection, when the detection area is located at 0.3 times the width of the measured object, in addition to considering the influence of the internal defects of concrete and the dominant frequency of material degradation, it is also necessary to consider the influence of the component boundary on the detection results.

(3) Through the plane strain finite element simulation, the dominant frequency at the edge is 0.9 times the thickness frequency. The dominant frequency is the natural frequency corresponding to the specific mode under the cross-section. When the distance between the acquisition point and the edge of the concrete column becomes larger, the influence of the edge is also reduced and the dominant frequency is restored to the thickness frequency corresponding to Mode 2.

**Author Contributions:** Conceptualization, writing—review and editing, Y.L.; writing—original draft preparation, methodology and software, H.X.; formal analysis and validation, X.M.; project administration, D.W.; supervision, X.H. All authors have read and agreed to the published version of the manuscript.

**Funding:** This research was funded by Anhui Provincial Universities Natural Science Research Project, grant number KJ2020ZD43; Anhui Housing Urban-Rural Development Scientific and Technological projects (2022-RK057); Research Fund Project of Anhui Jianzhu University (2019QDZ51); and Anhui Provincial Universities Natural Science Research Project (KJ2021A0607).

**Institutional Review Board Statement:** Not applicable.

**Informed Consent Statement:** Not applicable.

**Data Availability Statement:** The datasets included in and/or analyzed during the current study are available from the corresponding author upon reasonable request.

**Acknowledgments:** The authors are grateful to the Prefabricated Building Research Institute of Anhui Province for funding and providing research facilities for this study.

**Conflicts of Interest:** The authors declare no conflict of interest.

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
