# Peer review of "Study on Impact–Echo Response of Concrete Column near the Edge"

_applsci, doi:10.3390/app13095590_

Round 1

Reviewer 1 Report

Dear Authors,

The structural dynamic model needs better mathematical details. The topic looks very generalized, must be brought into particular domain. My observations are as follows:p

1. Which specific impact/domain your model is applicable for?

2. How exactly you define, NEAR THE EDGE? How

3. talking about bigger domain BEM has its applications ofcourse, looks like ref 9 belongs to the similar research group. It must be specified with the novel parameters otherwise the degree of novelty becomes quite low.

4. Introduction needs better referencing apart from the particular group.

5. Numerical model/principle (section 2) needs complete revision. Please properly mention the boundary conditions and methods properly.

6.  Cite the equations in text and write techynically. It looks non-professional when it comes to mathematics.

7.  Figure quality overall is low and not well placed. 

8. The comparison and validation part needs better presentation.

Try to consult some similar study and explaning figures/tables/equations need proper English. You will get an idea.

Reviewer 2 Report

Dear authors,

my comments are in the attached file.

Best regards

Dear authors,

my comments regarding style and English are in attached file as well.

Best regards

Round 2

Reviewer 1 Report

The points are responded perfectly.

The writing is improved and qualifies the criteria.

Reviewer 2 Report

Dear authors,

I have went through your revised paper and notes, and I conclude, that you have addressed all my remarks. I appreciate that you have added photos of actual used equipment and photos from the in-situ measurements. If you can not present the mixture of used concrete, please add what type of concrete it was at least by its strength grade.

I agree with your revised paper and it can be published. Congrats!